# Impact of pyrazinamide usage on serious adverse events in elderly tuberculosis patients: A multicenter cohort study

Joon Young Yoon[1], Tae-Ok Kim[1], Ju Sang Kim[2], Hyung Woo Kim[2], Eung Gu Lee[3], Sung Soo Jung[4], Jee Youn Oh[5], Jin Woo Kim[6], Sang Haak Lee[7], Seunghoon Kim[8], Sun-Hyung Kim[9], Yeonhee Park[10], Jinsoo Min[11]*, Yong-Soo Kwon[1]*

1 Department of Internal Medicine, Chonnam National University Hospital, Chonnam National University Medical School, Gwangju, South Korea, 2 Division of Pulmonary and Critical Care Medicine, Department of Internal Medicine, Incheon St. Mary's Hospital, College of Medicine, The Catholic University of Korea, Seoul, Republic of Korea, 3 Division of Pulmonary, Allergy, and Critical Care Medicine, Department of Internal Medicine, Bucheon St. Mary's Hospital, College of Medicine, The Catholic University of Korea, Seoul, Republic of Korea, 4 Division of Pulmonary and Critical Care Medicine, Department of Internal Medicine, Chungnam National University Hospital, Daejeon, Republic of Korea, 5 Division of Pulmonary, Allergy, and Critical Care Medicine, Department of Internal Medicine, Korea University Guro Hospital, Korea University College of Medicine, Seoul, Republic of Korea, 6 Division of Pulmonary and Critical Care Medicine, Department of Internal Medicine, Uijeongbu St. Mary's Hospital, College of Medicine, The Catholic University of Korea, Seoul, Republic of Korea, 7 Division of Pulmonary, Critical Care, and Sleep Medicine, Department of Internal Medicine, Eunpyeong St. Mary's Hospital, College of Medicine, The Catholic University of Korea, Seoul, Republic of Korea, 8 Division of Pulmonary and Critical Care Medicine, Department of Internal Medicine, St. Vincent's Hospital, College of Medicine, The Catholic University of Korea, Seoul, Republic of Korea, 9 Division of Pulmonary and Critical Care Medicine, Department of Internal Medicine, Chungbuk National University Hospital, Cheongju, Republic of Korea, 10 Division of Pulmonary and Critical Care Medicine, Department of Internal Medicine, Daejeon St. Mary's Hospital, College of Medicine, The Catholic University of Korea, Seoul, Republic of Korea, 11 Division of Pulmonary and Critical Care Medicine, Department of Internal Medicine, Seoul St. Mary's Hospital, College of Medicine, The Catholic University of Korea, Seoul, Republic of Korea

☯ These authors contributed equally to this work.
* minjinsoo@catholic.ac.kr (JM); yskwon@jnu.ac.kr (YSK)

**Data Availability Statement:** The data that support the findings of this study are available from Korea Disease Control and Prevention Agency, however, restrictions apply. Data will be provided by the

## Abstract

### Background

Pyrazinamide (PZA) usage has been associated with adverse drug reactions, prompting its avoidance in treating elderly tuberculosis (TB) patients. This study aims to examine whether the administration of PZA is associated with poor outcomes during TB treatment among elderly individuals.

### Methods

A retrospective analysis was undertaken on data collected from a prospective cohort conducted between July 2019 and June 2023, which involved tuberculosis patients from 18 institutions across the Republic of Korea. The study aimed to assess the impact of PZA on the incidence of serious adverse events (SAEs), medication interruptions, and becoming loss to follow-up (LTFU) during standard short courses of TB treatment in elderly ($\geq$65 years old) patients.

Korea Disease Control and Prevention Agency (Tae-hyoun Kim, D.V.M., Ph.D. whitevet81@korea. kr, Division of Bacterial Disease Research, Center for Infectious Disease Research, National Institute of Health, Korea Disease Control and Prevention Agency, Cheongju, South Korea) upon reasonable request.

**Funding:** This study was supported by a grant (BCRI24033) from Chonnam National University Hospital Biomedical Research Institute, awarded to YSK, and the Research Program funded by the Korea National Institute of Health (grant number 2022E200100 & 2020ER520502), awarded to JM. The funders had no role in study design, data collection, analysis, the decision to publish, or preparation of the manuscript.

**Competing interests:** The authors have declared that no competing interests exist.

## Results

PZA was administered to 356 of 390 elderly patients (91.3%), and 98 of the 390 (25.1%) experienced SAEs. Treatment success was significantly lower in patients not treated with PZA compared to those who received PZA (64.7% vs 89.9%, $p < 0.001$). The incidence of SAEs, medication interruption, or LTFU was higher in patients not given PZA compared those who received PZA (52.9% vs. 27.2%, $p = 0.002$). A multivariate logistic regression analysis, factoring in covariates such as age, comorbidities, and baseline laboratory data, revealed that PZA was not a risk factor for SAEs, medication interruption, or LTFU in TB treatment (odds ratio [OR] 0.457, 95% confidence interval [CI] 0.201–1.041).

## Conclusion

Treating elderly TB patients with PZA did not increase the incidence of SAEs, medication interruptions, or LTFU during the standard short course of TB treatment. Therefore, considering its potential advantages, incorporating PZA into the treatment regimen for elderly TB patients may be advisable.

## Introduction

Tuberculosis (TB) is both preventable and curable, yet it persists as the primary cause of death among infectious diseases globally, with more than 10 million individuals falling ill with TB annually [1]. Notably, there is a growing proportion of elderly individuals affected by TB [2–4].

Current guidelines for TB treatment recommend a short course of chemotherapy for drug-susceptible TB, including isoniazid (INH), rifampin (RIF), ethambutol (EMB), and pyrazinamide (PZA), a regimen known as HREZ. PZA is recommended to intensify the regimen during the first 2 months of treatment [5]. However, adverse drug reactions are reported to be more common with PZA than other drugs [6–8]. Because adverse events can lead to medication interruptions, such patients are more vulnerable to treatment failure [9]. Adverse events associated with TB medication are notably more prevalent among elderly patients [10–14]. Consequently, some guidelines and experts caution against the use of PZA in this demographic [5, 11]. However, despite these concerns, PZA exhibits bactericidal activity against quiescent bacilli, thereby reducing the risk of relapse [15]. Several studies have suggested that the use of PZA in elderly patients does not result in a higher incidence of side effects compared to INH, EMB, and RIF treatment alone (a regimen known as HRE) [16–18]. Notably, the majority of these studies were conducted in Japan. A need remains to gather additional evidence regarding the tolerability of PZA in elderly TB patients. The aim of our study was to assess HREZ and HRE regimens to determine whether PZA usage is associated with a higher incidence of serious adverse events (SAEs), treatment interruptions, or loss to follow-up (LTFU).

## Methods

### Study design and population

Data was extracted from the Cohort Study of Pulmonary Tuberculosis, a prospective cohort study [19]. The cohort consisted of 1,204 adult patients (aged $\geq$ 19) diagnosed with pulmonary TB, who agreed to participate in the study conducted at 18 university-affiliated hospitals in the Republic of Korea between August 2019 and December 2021. The Republic of Korea is an

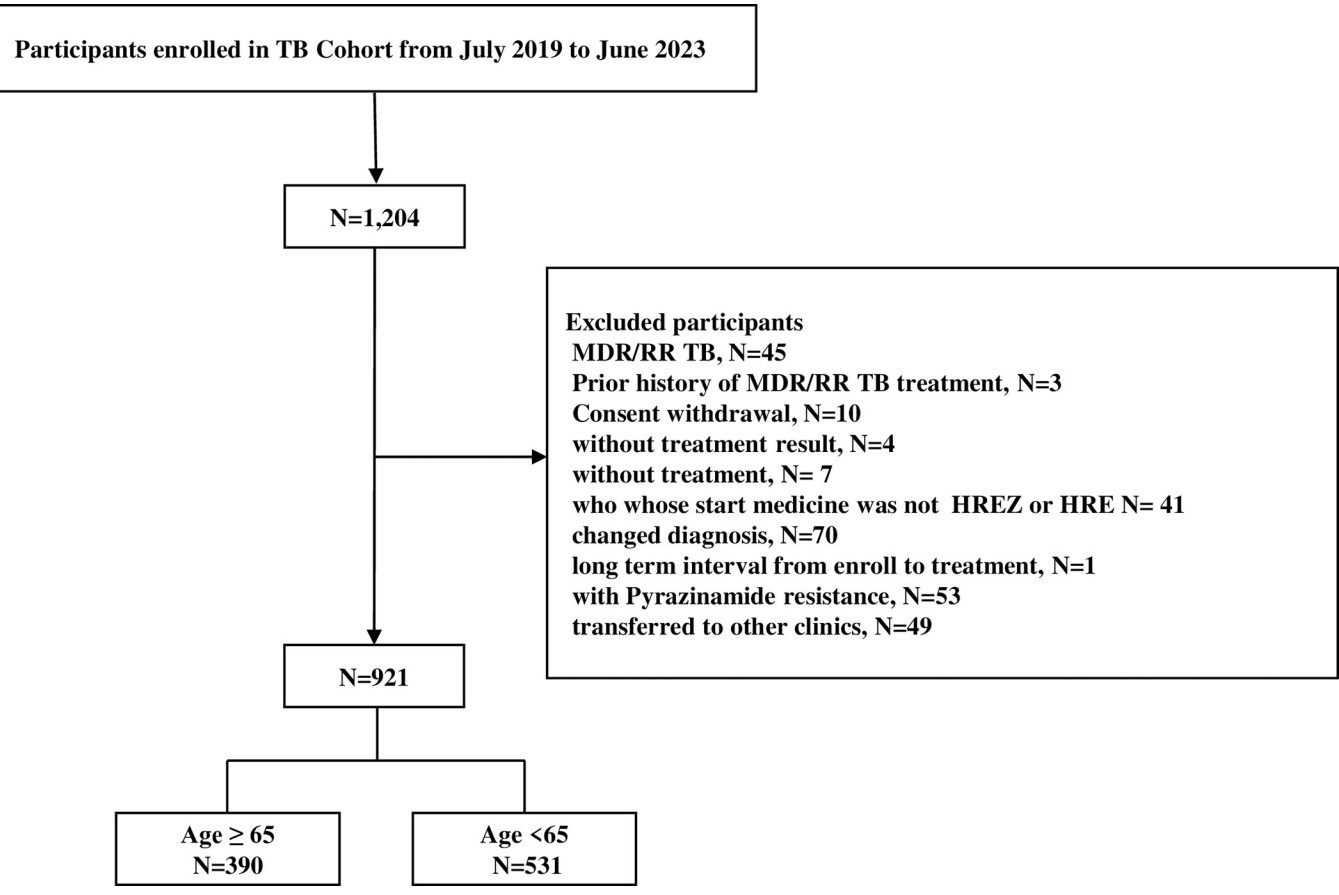

**Fig 1. Flowchart of participant enrollment.**

intermediate TB country with its high incidence among elderly populations [20]. In 2023, there were 13,285 reported cases of TB among the elderly (age ≥ 65), with a significantly higher rate of 104 cases per 100,000 population compared to the overall rate of 38.2 cases per 100,000 population [21].

We applied the following exclusion criteria to refine our analysis: (1) patients with multi-drug-resistant (MDR)- or rifampicin-resistant (RR)-TB (n = 45), (2) individuals with a history of prior treatment for MDR/RR-TB (n = 3), (3) individuals who did not receive treatment (n = 7), (4) those lacking treatment results (n = 49), (5) patients whose initial medication regimen was not HREZ or HRE (n = 41), (6) patients with pyrazinamide resistance (n = 53). (**Fig 1**).

## Follow up and data collection

Baseline demographic variables, such as gender, age, body mass index, smoking status, and comorbidities, were recorded for all participants. Each patient underwent comprehensive mycobacterial assessments and initial biochemical tests. Throughout the follow-up period, participants received regular biochemical evaluations and were closely monitored for adverse drug reactions at predefined intervals (14 days, 28 days, 2 months, and monthly thereafter). Follow-up continued until the completion or discontinuation of treatment, death, or the final hospital visit [19].

## Groups

An "elderly patient" was defined as ≥65 years old at the time of TB diagnosis; any patient <65 years old was considered a "young patient." A patient >75 years was defined as a "very old patient." Patients were also stratified based on the PZA usage. The cohort called "without PZA" was defined as follows: (1) receiving the HRE regimen, or (2) being prescribed PZA for less than 28 days, not due to adverse drug reactions (ADRs) but rather because clinicians were concerned about potential side effects given the patient's condition. In contrast, the cohort called "with PZA" was defined as follows: (1) opting for a treatment regimen including PZA for 28 days or longer, or (2) experiencing a SAE during the period of PZA usage, even if the duration was less than 28 days.

## Study outcomes

The primary outcomes were defined as follows: (1) SAE, (2) Medication interruption (cessation of all TB medication for a duration of 7 days or more), and (3) LTFU. The definition of a SAE encompassed any of the following criteria: (i) death, (ii) life-threatening situations, (iii) cases necessitating hospitalization or prolongation of hospital stay, (iv) instances leading to sustained or significant disability or impairment, (v) other significant medical events, and (vi) severe ADRs of severity grade 3 or higher [19]. A single patient can experience multiple SAEs, and each SAE was recorded.

Treatment outcomes for drug-sensitive TB were categorized in accordance with the Korean National TB outcome definitions: "cured" (a patient with bacteriologically confirmed TB exhibiting negative smear or culture results in the last month of treatment and on at least one other previous occasion); "completed" (a patient who finishes treatment without evidence of failure but has no recorded smears or culture results); "treatment failure" (a patient with TB showing a positive sputum smear or culture at month 5 or later during treatment); "died" (a patient with TB who dies for any reason before or during the course of treatment; "LTFU" (a patient with TB who does not start treatment or whose treatment is interrupted for two consecutive months or more). "Treatment success" is the sum of cured and completed cases.

## Covariates

Re-treatment indicates that the patient had previously been treated for TB. Chest X-rays or CT scans assessed whether patients had multi-lobe infiltration or cavitary lesions. Acid-fast bacilli (AFB) smears during the initial examination were used to assess AFB-positive status in patients with pulmonary TB. The definition of chronic pulmonary disease in this study includes patients with asthma, chronic bronchitis, emphysema, and other chronic lung disease who have ongoing symptoms such as dyspnea or cough, with mild or moderate activity. Baseline blood sampling was conducted before the initiation of treatment to establish initial patient parameters: anemia (hemoglobin <13 g/dL for men and <12 g/dL for women); hypoalbuminemia (albumin levels <3.5 g/dL); hyperbilirubinemia (total bilirubin level >1.2 mg/dL); abnormal liver function (aspartate aminotransferase [22] >40 IU/L or alanine aminotransferase [9] >40 IU/L); elevated serum creatinine (>1.2 mg/dL).

## Statistical analysis

Data are presented as means ± standard deviations for normally distributed continuous variables, and as medians ± interquartile ranges (IQRs) for non-normally distributed numbers with proportion for categorical variables. Continuous variables were analyzed using the student's t-test or Mann–Whitney U test. Categorical variables were analyzed using Pearson's

chi-square test or Fisher's exact test. Predictors for the occurrence of primary outcomes were selected based on demographic characteristics, comorbidities, radiologic findings, acid-fast bacilli smear results, and laboratory findings of the patients. Univariate logistic regression was performed first to estimate the association between a predictor and the occurrence of adverse results. Multivariate logistic regression was performed by including predictors exhibiting significant differences with P-values of <0.1 in the univariate logistic regression. P-values of <0.05 were considered statistically significant. All data analyses were conducted using IBM SPSS Statistics version 25 (SPSS Inc., Chicago, IL, USA).

### Ethics statement

This study was performed in accordance with the Declaration of Helsinki and was approved by Institutional Review Board of the Catholic University of Korea (IRB No. C19ONDI0458). All adult participants provided written informed consent to participate in the cohort study of pulmonary tuberculosis.

## Results

### Baseline characteristics

Among 390 elderly patients, 34 (8.7%) were categorized as without PZA. Patients without PZA were older (79.35 ± 7.18 years vs. 75.13 ± 7.01 years, p = 0.001), less likely to be male (35.3% vs. 59.3%, p = 0.007), and had lower blood albumin levels (3.34 ± 0.74g/dL vs. 3.75 ± 0.63g/dL, p = 0.005) compared to patients categorized as with PZA (**Table 1**).

### Study outcomes

The without PZA cohort had a lower treatment success rate (64.7% vs. 89.9%, p < 0.001) and longer treatment duration (272 [IQR 230–280] days vs. 189 [IQR 182–245] days, p < 0.001) than the with PZA cohort (Table 2). Primary outcomes (SAE, medication interruption, or LTFU) occurred significantly less frequently in with PZA patients than without PZA patients (**Table 2**). This trend was also observed in patients aged 75 years and older. (**S1 Table**).

  Elderly TB patients had a lower treatment success rate and more frequent primary outcomes than young TB patients (**S2 Table**).

  In the analysis of patients treated with PZA based on the presence of SAEs, those with SAEs were older, had more multi-lobe infiltration, and had lower serum albumin levels at baseline. Regarding treatment outcomes, patients with SAEs had a lower treatment success rate and a higher rate of medication interruption. (S3 Table).

### Risk factors for primary outcomes

In univariate analyses, very old age (≥75 years), PZA usage, underlying chronic pulmonary disease, anemia, hypoalbuminemia, and elevated serum creatinine were significantly associated with primary outcomes. In the multivariate analysis, chronic pulmonary disease, anemia, and elevated serum creatinine were independent risk factors for primary outcome. However, PZA usage was not a significant factor in this analysis (**Table 3**).

## Discussion

Treating TB in elderly patients poses challenges because of more frequent SAEs than young patients, with reported rates ranging from 10% to 30% [7, 10–13]. Furthermore, multiple researchers have stated that PZA may be a significant contributor to ADRs in elderly TB patients [7, 10, 12, 14, 23]. Consequently, clinicians may hesitate to incorporate PZA into

**Table 1. Baseline characteristics of elderly tuberculosis patients based on pyrazinamide usage.**

| Variables | Total, n = 390 | Without PZA, n = 34 (8.7%) | With PZA, n = 356 (91.3%) | *P* value |
|---|---|---|---|---|
| Age, years | 75.49 ± 7.12 | 79.35 ± 7.18 | 75.13 ± 7.01 | 0.001 |
| Males, n (%) | 223 (57.2) | 12 (35.3) | 211 (59.3) | 0.007 |
| BMI, kg/m$^2$ | 21.87 ± 3.37 | 22.56 ± 3.98 | 21.98 ± 3.25 | 0.329 |
| Ever smoker (%) | 175 (44.9) | 11 (32.4) | 164 (46.1) | 0.124 |
| Re-treatment. n/N (%) | 65/384 (16.9) | 6/33 (18.2) | 59/351 (16.8) | 0.841 |
| Extrapulmonary TB, n (%) | 39 (10) | 5 (14.7) | 34 (9.6) | 0.364 |
| Multi-lobe infiltration, n/N (%) | 344/383 (89.8) | 33/34 (97.1) | 311/349 (89.1) | 0.231 |
| Cavitary lesion, n/N (%) | 46/379 (12.1) | 2/34 (5.9) | 44/345 (12.8) | 0.406 |
| AFB Smear positive, n/N (%) | 214/306 (69.9) | 16/23 (69.6) | 198/283 (70) | 0.968 |
| Chronic pulmonary disease, n (%) | 35 (9) | 6 (17.6) | 29 (8.1) | 0.105 |
| Renal disease, n (%) | 18 (4.6) | 4 (11.8) | 14 (3.9) | 0.061 |
| Liver disease, n (%) | 9 (2.3) | 1 (2.9) | 8 (2.2) | 1.000 |
| Cancer, n (%) | 47 (12.1) | 5 (14.7) | 42 (11.8) | 0.783 |
| Hemoglobin, g/dL[a] | 12.24 ± 2.87 | 11.74 ± 1.60 | 12.30 ± 2.91 | 0.294 |
| Albumin, g/dL[b] | 3.71 ± 0.65 | 3.34 ± 0.74 | 3.75 ± 0.63 | 0.005 |
| Total bilirubin, mg/dL [c] | 0.64 ± 0.51 | 0.98 ± 1.22 | 0.59 ± 0.28 | 0.086 |
| AST, IU/L [d] | 28.57 ± 25.35 | 41.71 ± 49.71 | 26.80 ± 18.70 | 0.107 |
| ALT, IU/L [e] | 20.47 ± 17.83 | 23.61 ± 24.85 | 19.94 ± 15.96 | 0.426 |
| Creatinine(mg/dL) [f] | 1.06 ± 0.56 | 1.00 ± 0.51 | 0.89 ± 0.55 | 0.262 |

Abbreviations: AFB: Acid-Fast Bacillus; ALT: alanine aminotransferase; AST: aspartate aminotransferase; BMI: Body mass index; TB: Tuberculosis

[a] Total n = 354; without PZA n = 31; with PZA n = 326

[b] Total n = 344; without PZA n = 31; with PZA n = 313

[c] Total n = 343; without PZA n = 31; with PZA n = 312

[d] Total n = 353; without PZA n = 31; with PZA n = 322

[e] Total n = 353; without PZA n = 31; with PZA n = 322

[f] Total n = 355; without PZA n = 31; with PZA n = 324

initial treatment regimens, and some experts advocate for regimens that exclude PZA during the intensive treatment phase for elderly TB patients [5]. However, other reports state that PZA does not significantly increase the risk of ADRs, indicating a lack of consensus on this matter [11, 16, 23]. Importantly, data comparing the efficacy of regimens with and without PZA in elderly TB patients are limited. In this study, most elderly patients (91.2%) received PZA in their initial regimen, and it did not raise the frequency of ADRs. Instead, our data suggest that PZA in an initial regimen may improve treatment outcomes in elderly TB patients. The disparity between our study and that reported more frequent ADRs in elderly TB patients receiving PZA in initial regimens could be due to variations in population demographics, including comorbidities, access to medical and socioeconomic resources, and differences in the management of ADRs.

Patients who were LTFU could have higher SAE incidences, including multidrug-resistant TB development and mortality, and repeated LFTU [24–26]. Importantly, intolerances to TB medications could be a risk factor for incomplete treatment [18]. Therefore, ADRs and SAEs can lead to LTFU outcomes in TB patients [22, 27] and increase the incidence of unsuccessful treatment outcomes in elderly TB patients. However, in our study, use of PZA did not increase the frequency of LTFU or primary outcomes, including SAEs, medication interruption, and LTFU compared to those in patients without PZA. Interestingly, our data showed that PZA usage was associated with better primary outcomes and treatment success. Considering

**Table 2. Treatment success and primary outcomes in elderly tuberculosis patients based on pyrazinamide usage.**

| Variables | Total, n = 390 | Without PZA, n = 34 (8.7%) | With PZA, n = 356 (91.3%) | P value |
|---|---|---|---|---|
| Treatment Success, n (%) | 342 (87.7) | 22 (64.7) | 320 (89.9) | <0.001 |
| PZA use time, median days (IQR) | 61 (55–66) | 0 (0–7) | 62 (56–67) | <0.001 |
| Treatment duration, median days (IQR) | 190 (182–254) | 272 (230–280) | 189 (182–245) | <0.001 |
| SAEs, n (%) | 98 (25.1) | 14 (41.2) | 84 (23.6) | 0.024 |
| Time to first SAE, median days (IQR) | 31 (14–74) | 28 (21–52) | 32 (14–85) | 0.994 |
| SAEs Category | | | | |
| Hepatotoxicity, n (%) | 30 (7.7) | 4 (11.8) | 26 (7.3) | 0.317 |
| Generalized weakness, n (%) | 17 (4.4) | 11 (32.4) | 6 (1.7) | <0.001 |
| Cytopenia, n (%) | 12 (3.1) | 5 (14.7) | 7 (2.0) | 0.002 |
| Infection, n (%) | 12 (3.1) | 0 (0) | 12 (3.4) | 0.611 |
| Dyspnea, n (%) | 11 (2.8) | 1 (2.9) | 10 (2.8) | 1.000 |
| Gastrointestinal adverse drug reactions, n (%) | 10 (2.6) | 1 (2.9) | 9 (2.5) | 0.603 |
| Cutaneous adverse drug reactions, n (%) | 9 (2.3) | 1 (2.9) | 8 (2.2) | 0.564 |
| Visual defect, n (%) | 5 (1.3) | 1 (2.9) | 4 (1.1) | 0.368 |
| Others, n (%) | 30 (7.7) | 0 (0) | 30 (8.4) | 0.094 |
| Death, n (%) | 35 (9) | 9 (26.5) | 26 (7.3) | 0.001 |
| Medication interruption, n (%) | 38 (10.67) | 4 (1.12) | 34 (9.55) | 0.760 |
| LTFU, n (%) | 6 (1.69) | 2 (0.56) | 4 (1.12) | 0.089 |
| SAEs, medication interruption, or LTFU, n (%) | 115 (29.5) | 18 (52.9) | 97 (27.2) | 0.002 |

Abbreviations: IQR: Interquartile range; LTFU: lost to follow-up; PZA: Pyrazinamide; SAE: Serious adverse event

treatment duration, the PZA-containing regimen (HREZ) is much shorter than the regimen without PZA (HRE), with durations of 6 months versus 9 months, respectively. Therefore, patients receiving PZA had a shorter treatment duration, which could result in less adverse drug reactions, medication interruption, and instances of LTFU, leading to better outcomes.

One study reported that patients who received a non-standard initial regimen for TB treatment required longer treatment and experienced more frequent treatment interruptions [28]. PZA was the most common drug omitted from standard regimens, and the risk factors associated with non-standard initial regimens were underlying diseases, including eye disease, liver disease, gout, or hyperuricemia. Physicians' concerns about patients with such underlying diseases developing ADRs could be one reason for prescribing non-standard initial regimens without PZA. In our study, we did not collect data about the reasons for not prescribing PZA in an initial regimen and co-morbid conditions, including liver disease, were not different between the two groups, but it is possible that attending physicians' concern about development of ADRs was a factor in omitting PZA in the initial regimen, because patients without PZA were older and had lower serum albumin levels, which might have been indicative of poorer nutritional status, than patients with PZA. Additionally, patients without PZA had higher death rates and generalized weakness as a SAE compared to those with PZA. Thus, patients without PZA might have had poorer overall health than those with PZA. However, due to the retrospective design of this study, we could not ascertain the exact health conditions of the enrolled patients, which is a limitation of our study.

The hepatotoxicity of PZA can be a major concern, especially in elderly patients. In the earlier study, PZA-induced hepatotoxicity was frequent in elderly TB patients [29]. In our study, there was no difference in hepatotoxicity between patients with and without PZA. The discrepancy could be due to differences in the definition of hepatotoxicity. In our study, we only scored hepatotoxicity that presented as a SAE. Differences in enrolled patients included

**Table 3. Risk factors for the occurrence of severe adverse reactions, medication interruption, or becoming lost to follow-up.**

| Variables | Univariate | | | Multivariate | | |
|---|---|---|---|---|---|---|
| | OR | 95%CI | P value | OR | 95%CI | P value |
| Age ≥75 | 1.84 | 1.18–2.86 | .007 | 1.16 | 0.69–1.95 | .580 |
| Gender, Male | 1.11 | 0.72–1.72 | .639 | | | |
| PZA use | 0.30 | 0.15–0.61 | .001 | 0.46 | 0.20–1.04 | .062 |
| Comorbidities | | | | | | |
| Chronic pulmonary disease | 2.74 | 1.36–5.53 | .005 | 2.85 | 1.28–6.36 | .010 |
| Liver disease | 0.66 | 0.14–3.23 | .609 | | | |
| Cancer | 0.94 | 0.49–1.79 | .845 | | | |
| BMI <18.5 kg/m$^2$ | 0.84 | 0.44–1.62 | .601 | | | |
| Ever smoker | 1.31 | 0.85–2.02 | .222 | | | |
| Re-treatment | 1.16 | 0.65–2.06 | .612 | | | |
| Concurrent extrapulmonary TB | 1.72 | 0.87–3.39 | .117 | | | |
| Multi-lobar infiltration | 2.12 | 0.91–4.96 | .083 | 1.49 | 0.60–3.73 | .393 |
| Cavitary lesion | 0.79 | 0.39–1.58 | .504 | | | |
| AFB smear positive | 0.80 | 0.48–1.33 | .378 | | | |
| Anemia[b] | 3.13 | 1.89–5.19 | .000 | 2.75 | 1.61–4.71 | .000 |
| Hypoalbuminemia[c] | 2.35 | 1.45–3.79 | .000 | 1.30 | 0.75–2.27 | .354 |
| Hyperbilirubinemia[d] | 0.88 | 0.27–2.86 | .829 | | | |
| Abnormal liver function[e] | 1.51 | 0.78–2.90 | .218 | | | |
| Elevated serum creatinine[f] | 4.29 | 1.95–9.44 | .000 | 2.58 | 1.11–5.96 | .027 |

Abbreviations: AFB: acid-fast bacillus; BMI: body mass index; PZA: pyrazinamide; TB: tuberculosis

[b] Hemoglobin <13 g/dL for men, and < 12 g/dL for women

[c] Albumin <3.5 g/dL

[d] Total bilirubin >1.2 mg/dL

[e] Abnormal liver function tests defined as alanine aminotransferase >40 IU/L or aspartate aminotransferase >40 IU/L

[f] Serum creatinine >1.2 mg/dL

comorbidities, co-administered drugs, and ethnicity, in which differences in hepatic enzyme metabolization of TB drugs could also make a difference [30].

In our study, anemia, elevated serum creatinine, and the presence of chronic pulmonary disease were risk factors of SAEs including death, medication interruption, and LTFU. These findings are consistent with previous studies [8, 31–33]. In a study that evaluated TB patients with chronic kidney disease, ADRs were more frequent than in patients without chronic kidney disease, although statistical significance was marginal (p = 0.051) [31]. Anemia is a known a risk factor for death in TB patients [8, 32, 33]. Although previous studies did not report whether anemia was associated with more frequent ADRs, their findings could be consistent with our study because death was one of the main categories of SAEs in our study. COPD is also a known risk factor for death in TB patients [34–36].

Our study has several limitations. Firstly, it was not a randomized controlled trial, resulting in uneven baseline characteristics between the two groups. Patients in the without PZA group were older and had lower albumin levels, factors that could potentially influence the study outcomes. Secondly, the reasons behind clinicians' decisions to include or exclude PZA from initial TB treatment regimens were not documented. This lack of information introduces confounding variables that impact treatment outcomes, such as the overall health status of patients. Thirdly, some laboratory data, including hemoglobin, serum albumin, bilirubin, liver enzymes, and creatinine, were not available for all patients, which could affect the

completeness and accuracy of our analysis. Fourthly, the deaths in this study were due to all-cause mortality, not specifically TB-related deaths. Since the study enrolled elderly patients, and those without PZA were significantly older, TB-related death might be a more appropriate outcome measure than all-cause mortality for this study.

## Conclusion

Incorporating PZA into the initial regimen for TB treatment does not elevate frequency of adverse outcomes; instead, it may enhance treatment success rates. Consequently, in elderly patients with pulmonary TB, it is unnecessary to refrain from prescribing PZA as part of their TB treatment regimen.

## Supporting information

**S1 Table. Baseline characteristics and treatment results based on pyrazinamide usage in patients aged 75 years and older.**
(DOCX)

**S2 Table. Treatment success and primary outcomes based on age.**
(DOCX)

**S3 Table. Baseline characteristics and treatment outcomes based on the presence of severe adverse events in elderly tuberculosis patients with pyrazinamide.**
(DOCX)

## Author Contributions

**Conceptualization:** Joon Young Yoon, Tae-Ok Kim, Jinsoo Min, Yong-Soo Kwon.

**Data curation:** Joon Young Yoon, Tae-Ok Kim, Ju Sang Kim, Hyung Woo Kim, Eung Gu Lee, Sung Soo Jung, Jee Youn Oh, Jin Woo Kim, Sang Haak Lee, Seunghoon Kim, Sun-Hyung Kim, Yeonhee Park, Jinsoo Min, Yong-Soo Kwon.

**Formal analysis:** Joon Young Yoon, Tae-Ok Kim, Ju Sang Kim, Jinsoo Min, Yong-Soo Kwon.

**Funding acquisition:** Yong-Soo Kwon.

**Investigation:** Yong-Soo Kwon.

**Methodology:** Yong-Soo Kwon.

**Supervision:** Ju Sang Kim, Hyung Woo Kim, Eung Gu Lee, Sung Soo Jung, Jee Youn Oh, Jin Woo Kim, Sang Haak Lee, Seunghoon Kim, Sun-Hyung Kim, Yeonhee Park, Jinsoo Min.

**Writing – original draft:** Joon Young Yoon, Tae-Ok Kim, Yong-Soo Kwon.

**Writing – review & editing:** Ju Sang Kim, Hyung Woo Kim, Eung Gu Lee, Sung Soo Jung, Jee Youn Oh, Jin Woo Kim, Sang Haak Lee, Seunghoon Kim, Sun-Hyung Kim, Yeonhee Park, Jinsoo Min, Yong-Soo Kwon.

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
