## [Decision Letter · Decision Letter 0]

26 Jun 2024

PONE-D-24-12937Impact of Pyrazinamide Usage on Serious Adverse Events in Elderly Tuberculosis Patients: A Multicenter Cohort StudyPLOS ONE

Dear Dr. Kwon,

Thank you for submitting your manuscript to PLOS ONE. After careful consideration, we feel that it has merit but does not fully meet PLOS ONE’s publication criteria as it currently stands. Therefore, we invite you to submit a revised version of the manuscript that addresses the points raised during the review process. 

We look forward to receiving your revised manuscript.

Kind regards,

Lisa Kawatsu, PhD

Academic Editor

PLOS ONE

 [his study was supported by a grant (BCRI24033) from Chonnam National University Hospital Biomedical Research Institute. ].  

Additional Editor Comments (if provided):

Reviewers' comments:

Reviewer's Responses to Questions

**Comments to the Author**

1. Is the manuscript technically sound, and do the data support the conclusions?

Reviewer #1: Yes

Reviewer #2: Yes

2. Has the statistical analysis been performed appropriately and rigorously? 

Reviewer #1: Yes

Reviewer #2: Yes

3. Have the authors made all data underlying the findings in their manuscript fully available?

Reviewer #1: Yes

Reviewer #2: Yes

4. Is the manuscript presented in an intelligible fashion and written in standard English?

Reviewer #1: Yes

Reviewer #2: Yes

5. Review Comments to the Author

Reviewer #1: This is a retrospective observational study specifically in an important population group (the elderly) which is often neglected in clinical research. The study emphasizes the importance and benefits of using pyrazinamide regardless of age in treatment success for TB. It adds to the body of evidence to help guide treatment of TB in the elderly.

I have some comments and suggestions for the authors to address.

Line 120-121: please state the incidence of TB in the Republic of Korea and also among the elderly population.

Table 1:

a. What does overt smoker mean? Does overt smoker mean current smoker?

b. Does re-treatment mean that the patient had been previously treated for TB? This seems to be a high proportion of patients. Please explain.

Table 2:

Some of the terminology of SAEs are a little unusual. For example, with general weakness, do you mean generalized weakness? Please find alternatives for “gastrointestinal trouble”, “cutaneous trouble”.

Table 3: Suggest 2 decimal places for values. As chronic pulmonary disease has the highest OR in multivariate regression, you may need to explain the definition of chronic pulmonary disease.

For AFB-positive, is this only for pulmonary TB, or is extra-pulmonary TB smear included in this category?

Line 262: Was the duration of HREZ shorter than HRE because recommended treatment for HRE is 9 months compared with 6 months for HREZ? A shorter duration of treatment indicating a better outcome is a very generalized statement as other factors need to be considered such as treatment success, similar or reduced adverse events etc. Please rephrase.

Line 271: Could it be that the group not prescribed PYZ had poorer outcomes because they had poorer health in general? You implied this in line 274-275. 9 of 34 patients died in the “without PYZ” group and 11 of 34 had “general weakness” which appeared to be the main causes of SAEs.

Line 278: Your definition of death was all-cause mortality and not specifically TB-related death. For example, a patient may die with TB, not of TB. Also, given that the “without PZA” group was 4 years older than the “with PZA” group, this already puts them at a higher risk of death. Is there any data on the causes of death in your study?

Line 288: You mention presence of COPD but previously had written “chronic pulmonary disease”. Please clarify.

Line 292-293; line 295-296: When you refer to anemia and COPD being risk factors for death, does it refer to increased mortality risk specifically in TB disease?

It would be nice to see an analysis of the “with PYZ” group divided into those who had SAE vs those who did not have SAEs, given that the other primary outcomes of LTFU and treatment interruption does not seem to have had much overall impact. This would potentially be useful in finding out risk factors for SAEs specifically within the elderly population.

Minor comments:

Line 74: Republic of Korea

Line 76: loss to follow-up

Line 102: more vulnerable

Line 166: Change cavity lesion to cavitary lesion in keeping with Table 1

Reviewer #2: 1. The younger group (<65 years) shows lower risk of liver disfunction in comparison to >65 years old group as in the supplementary table 1. Does this valid both for HRZE group and HRE group? If this is valid only for HRZE group, it may be the reason to avoid PZA for the elderly.

2. Did you analyse for >75 years old group only for the risk factors of ADR? US guidelines says "Consequently, some experts avoid the use of PZA during the intensive phase among patients >75 years of age. "

6. PLOS authors have the option to publish the peer review history of their article (what does this mean?). If published, this will include your full peer review and any attached files.

Reviewer #1: **Yes: **Jin-Gun Cho

Reviewer #2: No

---

## [Author Response · Author response to Decision Letter 0]

2 Aug 2024

-> We checked style requirements. 

 [his study was supported by a grant (BCRI24033) from Chonnam National University Hospital Biomedical Research Institute. ]. 

-> We added funders and stated the role of the funders. 

-> We specifically outlined the data sharing policy of this study.

“The data that support the findings of this study are available from Korea Disease Control and Prevention Agency, however, restrictions apply. Data will be provided by the Korea Disease Control and Prevention Agency (Tae-hyoun Kim, D.V.M., Ph.D. whitevet81@korea.kr, Division of Bacterial Disease Research, Center for Infectious Disease Research, National Institute of Health, Korea Disease Control and Prevention Agency, Cheongju, South Korea) upon reasonable request..”

Reviewer #1: 

Reviewer #1: This is a retrospective observational study specifically in an important population group (the elderly) which is often neglected in clinical research. The study emphasizes the importance and benefits of using pyrazinamide regardless of age in treatment success for TB. It adds to the body of evidence to help guide treatment of TB in the elderly.

->Thank you for your helpful comments. 

I have some comments and suggestions for the authors to address.

Line 120-121: please state the incidence of TB in the Republic of Korea and also among the elderly population.

-> We added it as you recommended (Line 119). 

Table 1: a. What does overt smoker mean? Does overt smoker mean current smoker?

-> Thank you for your valuable comment. In our paper, the term 'overt smoker' includes both current and ex-smokers. To avoid confusion, we have replaced 'overt smoker' with 'ever smoker'.

b. Does re-treatment mean that the patient had been previously treated for TB? This seems to be a high proportion of patients. Please explain.

-> We understand that your definition of re-treatment aligns with ours. Re-treatment cases include patients who have previously undergone TB treatment, encompassing those with recurrence after treatment and treatment failure and interruptions during prior treatments.

Based on the nationwide data from the annual report on notified tuberculosis patients in Korea for 2023 by the Korea Disease Control and Prevention Agency (available at reference 21), re-treatment cases accounted for 13.3% (2,565 out of 19,540 patients) of all TB patients in South Korea. Considering the age of the enrolled patients in this study (elderly, age ≥ 65), we believe that the rate of re-treatment cases was not high compared to that in the nationwide data. 

Table 2: Some of the terminology of SAEs are a little unusual. For example, with general weakness, do you mean generalized weakness? Please find alternatives for “gastrointestinal trouble”, “cutaneous trouble”.

-> Thank you for your valuable comment. We changed terms to generalized weakness, gastrointestinal adverse drug reactions, and cutaneous adverse drug reactions. 

Table 3: Suggest 2 decimal places for values. As chronic pulmonary disease has the highest OR in multivariate regression, you may need to explain the definition of chronic pulmonary disease.

For AFB-positive, is this only for pulmonary TB, or is extra-pulmonary TB smear included in this category?

-> Thank you for your valuable comments. We adjusted the values to 2 decimal places in Table 3. 

The definition of chronic pulmonary disease in this study includes patients with asthma, chronic bronchitis, emphysema, and other chronic lung disease who have ongoing symptoms such as dyspnea or cough, with mild or moderate activity. We added it in the Methods section (Line 172).

For AFB-positive, this is only for pulmonary TB and we added it in the Methods section (Line 172)

Line 262: Was the duration of HREZ shorter than HRE because recommended treatment for HRE is 9 months compared with 6 months for HREZ? A shorter duration of treatment indicating a better outcome is a very generalized statement as other factors need to be considered such as treatment success, similar or reduced adverse events etc. Please rephrase.

-> Thank you for your valuable comments. We rephrase the sentence to “Considering treatment duration, the PZA-containing regimen (HREZ) is much shorter than the regimen without PZA (HRE), with durations of 6 months versus 9 months, respectively. Therefore, patients receiving PZA had a shorter treatment duration, which could result in less adverse drug reactions, medication interruption, and instances of LTFU, leading to better outcomes.” (Line 280)

Line 271: Could it be that the group not prescribed PYZ had poorer outcomes because they had poorer health in general? You implied this in line 274-275. 9 of 34 patients died in the “without PYZ” group and 11 of 34 had “general weakness” which appeared to be the main causes of SAEs.

-> Thank you for your valuable comments. We agreed with your comment that patients without PZA might have poorer health condition. However, due to retrospective design of this study, we did not know the exact health conditions of enrolled patients which could be a limitation of this study. We added it in the Discussion section (Line 296)

Line 278: Your definition of death was all-cause mortality and not specifically TB-related death. For example, a patient may die with TB, not of TB. Also, given that the “without PZA” group was 4 years older than the “with PZA” group, this already puts them at a higher risk of death. Is there any data on the causes of death in your study?

-> Thank you for your valuable comments. We agreed that TB-related death would be a better outcome measurement for this study. Unfortunately, we do not have data about causes of death in enrolled patients. We added it in the limitation of this study. 

Line 288: You mention presence of COPD but previously had written “chronic pulmonary disease”. Please clarify.

-> Thank you for your valuable comment. We change it to chronic pulmonary disease. 

Line 292-293; line 295-296: When you refer to anemia and COPD being risk factors for death, does it refer to increased mortality risk specifically in TB disease?

-> Anemia and COPD are indeed risk factors for death in TB patients, as shown in our reference papers. To clarify this, we have added "death in TB patients" to our manuscript.

It would be nice to see an analysis of the “with PYZ” group divided into those who had SAE vs those who did not have SAEs, given that the other primary outcomes of LTFU and treatment interruption does not seem to have had much overall impact. This would potentially be useful in finding out risk factors for SAEs specifically within the elderly population.

-> Thank you for your valuable comments. As you recommended, we analyzed patients with PZA based on the presence of SAEs. In terms of baseline characteristics, patients with SAEs were older, had more multi-lobe infiltration, and had lower serum albumin levels. Regarding treatment outcomes, patients with SAEs had a lower treatment success rate and a higher medication interruption rate. We have included these findings in the results section (line 234) and in Supporting Table 3.

Supporting Table 3. Baseline characteristics and treatment outcomes based on the presence of severe adverse events in elderly tuberculosis patients with pyrazinamide

Variables Total, n = 356 SAE,

n = 84 (23.6%) No SEA,

n = 272 (76.4%) P value

Baseline characters

Age, years 75.67 ± 7.21 76.99 ± 7.37 74.55 ± 6.81 0.005

Males, n (%) 211 (59.3) 50 (59.5) 161 (59.2) 0.957

BMI, kg/m2 21.97 ± 3.40 21.94 ± 3.04 21.99 ± 3.32 0.905

Ever smoker (%) 175 (44.9) 11 (32.4) 164 (46.1) 0.124

Re-treatment. n/N (%) 59/351 (16.8) 13/81 (16.0) 46/270 (17.0) 0.835

Extrapulmonary TB, n (%) 34 (9.6) 12 (14.3) 22 (8.1) 0.091

Multi-lobe infiltration, n/N (%) 311/349 (89.1) 79/83 (95.2) 232/266 (87.2) 0.042

Cavitary lesion, n/N (%) 44/345 (12.8) 9/83 (10.8) 35/262 (13.4) 0.549

AFB Smear positive, n/N (%) 198/283 (70.0) 49/74 (66.2) 149/209 (71.3) 0.413

Chronic pulmonary disease, n (%) 29 (8.1) 10 (11.9) 19 (7.0) 0.150

Renal disease, n (%) 14 (3.9) 6 (7.1) 8 (2.9) 0.083

Liver disease, n (%) 8 (2.2) 0 (0.0) 8 (2.9) 0.112

Cancer, n (%) 47 (13.2) 8 (9.5) 39 (14.3) 0.255

Hemoglobin, g/dLa 12.11 ± 1.77 12.17 ± 5.11 12.34 ± 1.70 0.767

Albumin, g/dLb 3.75 ± 0.63 3.51 ± 0.61 3.83 ± 0.62 <0.001

Total bilirubin, mg/dL c 0.58 ± 0.28 0.60 ± 0.30 0.58 ± 0.28 0.658

AST, IU/L d 26.94 ± 19.97 29.71 ± 30.66 25.86 ± 12.55 0.280

ALT, IU/L e 19.99 ± 16.64 20.32 ± 21.96 19.82 ± 13.50 0.812

Creatinine(mg/dL) f 0.90 ± 0.60 0.99 ± 0.58 0.85 ± 0.54 0.060

 Treatment Outcomes

Treatment Success, n (%) 320 (89.9) 58 (69.0) 262 (96.3) <0.001

Medication interruption, n (%) 34 (9.6) 23 (27.4) 11 (4.0) <0.001

LTFU, n (%) 5 (1.4) 1 (1.2) 4 (1.5) 0.849

Abbreviations: AFB: Acid-Fast Bacillus; ALT: alanine aminotransferase; AST: aspartate aminotransferase; BMI: Body mass index; IQR: interquartile range; LFTU: lost to follow-up; PZA: pyrazinamide; SAE: serious adverse event; TB: Tuberculosis

a Total n = 326; SAE n = 79; No SAE n = 247

b Total n = 313; SAE n = 78; No SAE n = 235

c Total n = 312; SAE n = 78; No SAE n = 234

d Total n = 322; SAE n = 79; No SAE n = 243

e Total n = 322; SAE n = 79; No SAE n = 243

f Total n = 324; SAE n = 79; No SAE n = 245

Minor comments:

Line 74: Republic of Korea

Line 76: loss to follow-up

Line 102: more vulnerable

Line 166: Change cavity lesion to cavitary lesion in keeping with Table 1

-> Thank you for your detailed comments. We changed these as you recommended. 

Reviewer #2: 1. The younger group (<65 years) shows lower risk of liver disfunction in comparison to >65 years old group as in the supplementary table 1. Does this valid both for HRZE group and HRE group? If this is valid only for HRZE group, it may be the reason to avoid PZA for the elderly.

-> We appreciate your detailed review comments. As shown in Table 2, there was no difference in hepatotoxicity between elderly TB patients receiving PZA and those not receiving PZA. Further statistical analysis confirmed that there were no differences in hepatotoxicity between the younger group or across all age groups.

Hepatotoxicity in all TB patients based on PZA usage

Variables Total, 

n = 921 Without PZA, 

n = 48 (5.2%) With PZA, 

n = 873 (94.8%) P value

Hepatotoxicity, n (%) 47 (5.1) 4 (8.3) 43 (4.9) 0.301

Hepatotoxicity in younger TB patients (<65 years) based on PZA usage

Variables Total, 

n = 531 Without PZA, 

n = 14 (2.6%) With PZA, 

n = 517 (97.4%) P value

Hepatotoxicity, n (%) 17 (3.2) 0 (0.0) 17 (3.3) 1.000

2. Did you analyse for >75 years old group only for the risk factors of ADR? US guidelines says "Consequently, some experts avoid the use of PZA during the intensive phase among patients >75 years of age. "

-> Thank you for your valuable comment. We analyzed our data as you recommended. Very old age group also showed same as elderly group in this study. We added it in the Result section (line 226) and supporting table 1. 

Supporting Table 1. Baseline characteristics and treatment results based on pyrazinamide usage in patients aged 75 years and older

Variables Total, n = 199 Without PZA,

n = 26 (13.1%) With PZA, 

n = 173 (86.9%) P value

Baseline characters

Age, years 81.62 ± 4.64 82.54 ± 4.58 81.14 ± 4.61 0.150

Males, n (%) 89 (44.7) 10 (38.5) 79 (45.7) 0.491

BMI, kg/m2 21.84 ± 3.61 22.61 ± 4.37 21.74 ± 3.37 0.366

Ever smoker (%) 62 (31.2) 8 (30.8) 54 (31.2) 0.964

Re-treatment. n/N (%) 28/197 (14.2) 4/25 (16.0) 24/172 (14.0) 0.784

Extrapulmonary TB, n (%) 24 (12.1) 5 (19.2) 19 (11.0) 0.229

Multi-lobe infiltration, n/N (%) 183/198 (92.4) 26/26 (100.0) 157/172 (91.3) 0.117

Cavitary lesion, n/N (%) 17/196 (8.7) 1/26 (3.8) 16/170 (9.4) 0.348

AFB Smear positive, n/N (%) 111/167 (66.5) 12/18 (66.7) 99/149 (66.4) 0.985

Chronic pulmonary disease, n (%) 22 (11.1) 6 (23.1) 16 (9.2) 0.036

Renal disease, n (%) 16 (8.0) 4 (15.4) 12 (6.9) 0.140

Liver disease, n (%) 3 (1.5) 1 (3.8) 2 (1.2) 0.344

Cancer, n (%) 21 (10.6) 3 (11.5) 18 (10.4) 0.861

Hemoglobin, g/dLa 11.57 ± 1.52 11.45 ± 1.62 11.53 ± 1.51 0.802

Albumin, g/dLb 3.57 ± 0.63 3.15 ± 0.71 3.62 ± 0.59 0.001

Total bilirubin, mg/dL c 0.65 ± 0.61 1.11 ± 1.36 0.58 ± 0.29 0.069

AST, IU/L d 20.26 ± 20.52 46.08 ± 55.71 27.78 ± 23.51 0.125

ALT, IU/L e 20.26 ± 20.52 24.13 ± 27.61 18.94 ± 18.16 0.230

Creatinine(mg/dL) f 0.96 ± 0.50 1.09 ± 0.52 0.93 ± 0.48 0.147

Treatment Results

Treatment Success, n (%) 165 (82.9) 16 (61.5) 149 (86.1) 0.002

SAEs, n (%) 62 (31.2) 11 (42.3) 51 (29.5) 0.188

Time to first SAE, median days (IQR) 35 (13-89) 28 (13-63) 46 (13-91) 0.693

Treatment duration, median days (IQR) 185 (179-217) 206 (20-180) 185 (180-203) 0.273

Medication interruption, n (%) 20 (10.1) 2 (7.7) 18 (10.4) 0.668

LTFU, n (%) 5 (2.5) 2 (7.7) 3 (1.7) 0.128

Abbreviations: AFB: Acid-Fast Bacillus; ALT: alanine aminotransferase; AST: aspartate aminotransferase; BMI: Body mass index; IQR: interquartile range; LFTU: lost to follow-up; PZA: pyrazinamide; SAE: serious adverse event; TB: Tuberculosis

a Total n = 182; without PZA n = 24; with PZA n = 158

b Total n = 178; without PZA n = 24; with PZA n = 154

c Total n = 179; without PZA n = 24; with PZA n = 155

d Total n = 181; without PZA n = 24; with PZA n = 157

e Total n = 181; without PZA n = 24; with PZA n = 157

f Total n = 181; without PZA n = 24; with PZA n = 157

---

## [Decision Letter · Decision Letter 1]

21 Aug 2024

Impact of Pyrazinamide Usage on Serious Adverse Events in Elderly Tuberculosis Patients: A Multicenter Cohort Study

PONE-D-24-12937R1

Dear Dr. Kwon,

We’re pleased to inform you that your manuscript has been judged scientifically suitable for publication and will be formally accepted for publication once it meets all outstanding technical requirements.

Kind regards,

Lisa Kawatsu, PhD

Academic Editor

PLOS ONE

Additional Editor Comments (optional):

Reviewers' comments:

Reviewer's Responses to Questions

**Comments to the Author**

1. If the authors have adequately addressed your comments raised in a previous round of review and you feel that this manuscript is now acceptable for publication, you may indicate that here to bypass the “Comments to the Author” section, enter your conflict of interest statement in the “Confidential to Editor” section, and submit your "Accept" recommendation.

Reviewer #1: All comments have been addressed

Reviewer #2: All comments have been addressed

2. Is the manuscript technically sound, and do the data support the conclusions?

Reviewer #1: Yes

Reviewer #2: Yes

3. Has the statistical analysis been performed appropriately and rigorously? 

Reviewer #1: Yes

Reviewer #2: Yes

4. Have the authors made all data underlying the findings in their manuscript fully available?

Reviewer #1: Yes

Reviewer #2: Yes

5. Is the manuscript presented in an intelligible fashion and written in standard English?

Reviewer #1: Yes

Reviewer #2: Yes

6. Review Comments to the Author

Reviewer #1: (No Response)

Reviewer #2: (No Response)

7. PLOS authors have the option to publish the peer review history of their article (what does this mean?). If published, this will include your full peer review and any attached files.

Reviewer #1: No

Reviewer #2: No

---

## [Editor Report · Acceptance letter]

17 Sep 2024

PONE-D-24-12937R1 

PLOS ONE

Dear Dr. Kwon, 

I'm pleased to inform you that your manuscript has been deemed suitable for publication in PLOS ONE. Congratulations! Your manuscript is now being handed over to our production team.

Kind regards, 

on behalf of

Dr. Lisa Kawatsu 

Academic Editor

PLOS ONE